# Hydrogen Sulfide-Evoked Intracellular Ca^2+^ Signals in Primary Cultures of Metastatic Colorectal Cancer Cells

**DOI:** 10.3390/cancers12113338

**Published:** 2020-11-11

**Authors:** Pawan Faris, Federica Ferulli, Mauro Vismara, Matteo Tanzi, Sharon Negri, Agnese Rumolo, Kostantinos Lefkimmiatis, Marcello Maestri, Mudhir Shekha, Paolo Pedrazzoli, Gianni Francesco Guidetti, Daniela Montagna, Francesco Moccia

**Affiliations:** 1Laboratory of General Physiology, Department of Biology and Biotechnology “L. Spallanzani”, University of Pavia, 27100 Pavia, Italy; faris.pawan@unipv.it (P.F.); sharon.negri01@universitadipavia.it (S.N.); 2Department of Biology, Cihan University-Erbil, 44001 Erbil, Iraq; 3Laboratory of Immunology Transplantation, Foundation IRCCS Policlinico San Matteo, 27100 Pavia, Italy; f.ferulli@smatteo.pv.it (F.F.); matteo.tanzi01@universitadipavia.it (M.T.); agnese.rumolo01@universitadipavia.it (A.R.); 4Laboratory of Biochemistry, Department of Biology and Biotechnology “L. Spallanzani”, University of Pavia, 27100 Pavia, Italy; mauro.vismara01@universitadipavia.it (M.V.); gianni.guidetti@unipv.it (G.F.G.); 5Department of Molecular Medicine, University of Pavia, 27100 Pavia, Italy; kostantinos.lefkimmiatis@unipv.it; 6Veneto Institute of Molecular Medicine, Foundation for Advanced Biomedical Research, 35131 Padua, Italy; 7Medical Surgery, Foundation IRCCS Policlinico San Matteo, 27100 Pavia, Italy; m.maestri@smatteo.pv.it; 8Faculty of Science, Department of Medical Analysis, Tishk International University-Erbil, 44001 Erbil, Iraq; mudhir.shekha@su.edu.krd; 9Medical Oncology, Foundation IRCCS Policlinico San Matteo, 27100 Pavia, Italy; p.pedrazzoli@smatteo.pv.it; 10Diagnostic and Pediatric, Department of Sciences Clinic-Surgical, University of Pavia, 27100 Pavia, Italy

**Keywords:** H_2_S, TRPV1, NCX, cancer, metastatic colorectal carcinoma, proliferation

## Abstract

**Simple Summary:**

Colorectal cancer (CRC) is the most common type of gastrointestinal cancer and the third most predominant cancer in the world. CRC is potentially curable with surgical resection of the primary tumor. The clinical problem of colorectal cancer, however, is the spread and outgrowth of metastases, which are difficult to eradicate and lead to a patient’s death. The failure of conventional treatment to significantly improved outcomes in mCRC has prompted the search for alternative molecular targets with the goal of ameliorating the prognosis of these patients. The present investigation revealed that exogenous delivery of hydrogen sulfide (H_2_S) suppresses proliferation in metastatic colorectal cancer cells by inducing an increase in intracellular Ca^2+^ concentration. H_2_S was effective on metastatic, but not normal, cells. Therefore, we propose that exogenous administration of H_2_S to patients affected by metastatic colorectal carcinoma could represent a promising therapeutic alternative.

**Abstract:**

Exogenous administration of hydrogen sulfide (H_2_S) is emerging as an alternative anticancer treatment. H_2_S-releasing compounds have been shown to exert a strong anticancer effect by suppressing proliferation and/or inducing apoptosis in several cancer cell types, including colorectal carcinoma (CRC). The mechanism whereby exogenous H_2_S affects CRC cell proliferation is yet to be clearly elucidated, but it could involve an increase in intracellular Ca^2+^ concentration ([Ca^2+^]_i_). Herein, we sought to assess for the first time whether (and how) sodium hydrosulfide (NaHS), one of the most widely employed H_2_S donors, induced intracellular Ca^2+^ signals in primary cultures of human metastatic CRC (mCRC) cells. We provided the evidence that NaHS induced extracellular Ca^2+^ entry in mCRC cells by activating the Ca^2+^-permeable channel Transient Receptor Potential Vanilloid 1 (TRPV1) followed by the Na^+^-dependent recruitment of the reverse-mode of the Na^+^/Ca^2+^ (NCX) exchanger. In agreement with these observations, TRPV1 protein was expressed and capsaicin, a selective TRPV1 agonist, induced Ca^2+^ influx by engaging both TRPV1 and NCX in mCRC cells. Finally, NaHS reduced mCRC cell proliferation, but did not promote apoptosis or aberrant mitochondrial depolarization. These data support the notion that exogenous administration of H_2_S may prevent mCRC cell proliferation through an increase in [Ca^2+^]_i_, which is triggered by TRPV1.

## 1. Introduction

Colorectal cancer (CRC) is the most common type of gastrointestinal cancer and the third most predominant cancer in the world. In 2018, around 1.8 million cases were reported by the World Health Organization (WHO) and 862,000 deaths were registered (WHO, 2018). These numbers are expected to increase by 80% in year 2035, reaching approximately 2.4 million cases and contributing to 1.3 million deaths worldwide [1]. CRC is potentially curable with surgical resection of the primary tumor [2]. The clinical problem of colorectal cancer, however, is the spread and outgrowth of metastases. Over the last decade, the development of new combinations of chemotherapeutic agents along with the introduction of targeted therapies improved survival of a cohort of metastatic CRC (mCRC) patients. Moreover, treatment of advanced disease is still associated with a poor prognosis and significant morbidity. The failure of conventional treatment to significantly improved outcomes in mCRC has prompted the search for alternative molecular targets with the goal of ameliorating the prognosis of these patients [2]. This makes of pivotal importance the search for alternative molecular targets with the goal of ameliorating the prognosis of these patients.

Exogenous administration of hydrogen sulfide (H_2_S) is emerging as an alternative anticancer treatment. H_2_S is the latest addition to the family of gasotransmitters, also including nitric oxide and carbon monoxide. H_2_S is endogenously generated from the precursor L-cysteine by pyridoxal-5’ phosphate-dependent (PLP) enzymes, including cystathionine β-synthase (CBS) and cystathionine γ-lyase (CSE), and 3-mercaptopyruvate sulfurtransferase (3MST) [3,4]. H_2_S is produced in response to appropriate cell stimulation and regulates a myriad of physiological processes, including vascular tone and blood flow regulation [5,6], angiogenesis [7], synaptic transmission [8], cellular stress, inflammation, apoptosis, and energy metabolism [4,9]. Not surprisingly, disruption of physiological H_2_S synthesis has been implicated in multiple disorders, including hypertension, Alzheimer’s disease, diabetes mellitus, ulcerative colitis, and end-stage renal disease [9,10], while high cellular or tissue levels of H_2_S are highly toxic and result in severe cytotoxic effects [11]. The double-edged role played by H_2_S has recently been highlighted in cancer. Endogenous production of H_2_S enhances tumor growth and metastasis by stimulating mitochondrial bioenergetics, by eliciting proliferation, migration and invasion, and by promoting angiogenesis [12,13,14]. Nevertheless, exogenous H_2_S administration through H_2_S-releasing compounds exerts a strong anticancer effect by suppressing cancer cell proliferation and/or inducing cancer cell apoptosis [12,14,15]. A recent series of studies revealed that exogenously delivered H_2_S suppressed proliferation in a panel of CRC cell lines, including HT-29, Caco-2, SW1116, HCT116, and DLD1 [16,17,18,19]. The mechanism whereby exogenous H_2_S affects CRC cell proliferation is yet to be clearly elucidated, although it could involve the expression of the cyclin-dependent kinase inhibitor p21^Cip^ [17] and an increase in intracellular Ca^2+^ concentration ([Ca^2+^]_i_) [20]. It has long been known that H_2_S is able to elevate [Ca^2+^]_i_ by inducing either Ca^2+^ release from the endoplasmic reticulum (ER) [7] or by promoting extracellular Ca^2+^ entry through multiple Ca^2+^ entry pathways [21,22], including Transient Receptor Potential Vanilloid 1 (TRPV1) [21,23,24,25], TRPV3, and TRPV6 [26], TRP Ankyrin 1 (TRPA1) [27,28], L- and T-type voltage-gated Ca^2+^ channels (VGCCs) [21,29,30]. Intracellular Ca^2+^ signaling plays a crucial role in CRC cell proliferation and migration [31,32,33,34,35]; however, it has been demonstrated that Ca^2+^-entry pathways which exert a mitogenic effect in commercial CRC cell lines fail to do so in primary cultures derived from CRC patients [36]. It would be, therefore, therapeutically relevant to assess the effect of exogenous H_2_S on patients-derived CRC cells.

TRPV1 is a polymodal non-selective cation channel, which is gated by multiple stimuli, including the dietary agonist capsaicin [37], noxious heat (>41 °C) [37], extracellular acidification [38], reactive oxygen species [39], and several vanillotoxins [40]. Furthermore, TRPV1 is regarded among the main mediators of H_2_S-induced extracellular Ca^2+^ entry in healthy cells [21,23,24,25]. Notably, exogenously delivered H_2_S may suppress cell proliferation by promoting TRPV1-mediated extracellular Ca^2+^ entry in multiple cancer cell lines [41], including leukemia, breast cancer, cervical carcinoma, whereas capsaicin-induced TRPV1 activation exerts an anticancer effect in CRC [42], breast cancer [43] and bladder cancer [44] cell lines. Therefore, the present investigation aimed at assessing for the first time whether and how exogenously added H_2_S exerts an anticancer effect in primary cultures of metastatic CRC (mCRC) cells. We focused on mCRC cells, such as mCRC, it is often not treatable and tends to develop resistance mechanisms towards conventional pharmacological therapies [2,45]. It is, therefore, mandatory to identify alternative strategies, which could be effectively be translated in the clinical practice, to eradicate mCRC cells. To do so, we exploited a multidisciplinary approach, including intracellular Ca^2+^ imaging, western blotting, immunocytochemistry, flow cytometer and pharmacological manipulation. Our findings demonstrated that H_2_S was able to trigger extracellular Ca^2+^ entry in mCRC cells by activating TRPV1 and the reverse (i.e., Ca^2+^ entry) mode of the Na^+^/Ca^2+^ exchanger (NCX). H_2_S-induced Ca^2+^ entry was in turn able to suppress mCRC proliferation by arresting the cell cycle in the S-phase, thereby confirming that exogenous delivery of H_2_S may represent a reliable strategy to treat metastatic CRC patients.

## 2. Results

### 2.1. H_2_S Evokes a Dose-Dependent Increase [Ca^2+^]_i_ in Primary Cultures of mCRC Cells

H_2_S was delivered to primary cultures of CRC cells loaded with the Ca^2+^-sensitive dye Fura-2 in the form of sodium hydrosulfide (NaHS), a water-soluble H_2_S donor that is widely employed to investigate H_2_S-induced intracellular Ca^2+^ signals in normal [46,47,48] as well as cancer [49] cells. Our preliminary experiments revealed that 100 µM NaHS evoked a robust increase in [Ca^2+^]_i_ in mCRC cells (Figure 1A), whereas it induced a significantly (*p* < 0.05) smaller Ca^2+^ response in primary CRC (pCRC) cells (Figure 1A,B) and in cells isolated from the adjacent non-neoplastic tissue, which was used as control (Ctrl) (Figure 1A,B). Similarly, NaHS-evoked intracellular Ca^2+^ signals were significantly (*p* < 0.05) larger in pCRC as compared to non-neoplastic cells (Figure 1A,B). As eradicating metastatic cells represents the therapeutic challenge to treat CRC [2,45] and the Ca^2+^ signals to exogenous H_2_S was remarkably lower in non-neoplastic cells and pCRC cells, we focused our attention on mCRC cells.

NaHS was found to evoke dose-dependent Ca^2+^ signals in mCRC cells. NaHS did not induce any discernible increase in [Ca^2+^]_i_ at concentrations lower than 5 µM, such as 2.5 µM (Figure 2A–C). The Ca^2+^ response to NaHS indeed appeared at 5 µM (Figure 2A,B), when the majority of mCRC cells produced a single Ca^2+^ transient in response to agonist stimulation (Figure 2A). A careful examination of the Ca^2+^ responses to increasing doses of NaHS revealed a U-shaped dose-response relationship, as previously reported in rat aortic endothelial cells [49]. Both the percentage of responding cells and the magnitude of the Ca^2+^ peak decreased as NaHS concentration raised from to 5 µM up to 50 µM and then increased again for a further elevation in NaHS dose (Figure 2B,C). Our analysis indicated that the highest Ca^2+^ response was induced by 100 µM NaHS, while there was no significant (*p* < 0.05) difference in the percentage of responding cells in the concentration range spanning from 75 µM to 300 µM (Figure 2B,C). In aggregate, these data suggest that 100 µM NaHS represent the most suitable dose to explore the mechanisms of H_2_S-induced intracellular Ca^2+^ signaling in mCRC. 

The kinetics of the Ca^2+^ response to 100 μM NaHS showed two main patterns even in cells from the same microscopic field. The most frequent pattern observed consisted in a rapid increase in [Ca^2+^]_i_ which rapidly decayed to the baseline on agonist removal (blue trace in Figure 3A). This transient increase in [Ca^2+^]_i_ was detected in ≈75% of the cells (Figure 3B). In the remaining 25% (Figure 3B), the initial Ca^2+^ peak elevation rapidly decayed to a sustained plateau phase that was maintained as long as NaHS was present in the bath (green trace in Figure 3A). We, finally, showed that the transient Ca^2+^ response to 100 µM NaHS was reversible after a short washout (Figure 3C).

### 2.2. The Ca^2+^ Response to H_2_S in Primary Cultures of mCRC Cells Depends on Extracellular Ca^2+^ Entry 

H_2_S elicits both endogenous Ca^2+^ release and extracellular Ca^2+^ entry in mammalian cells [7,21,22]. In order to assess the Ca^2+^ source recruited by NaHS to induce intracellular Ca^2+^ signals in mCRC cells, external Ca^2+^ was removed from the bath (0Ca^2+^). Under such conditions, the H_2_S donor failed to augment [Ca^2+^]_i_ in 138 out of 138 cells (100%) (Figure 4A–C), whilst it triggered a Ca^2+^ response in most cells subsequent to Ca^2+^ re-addition to the extracellular solution (Figure 4A–C). This finding strongly suggests that NaHS-induced Ca^2+^ signaling required Ca^2+^ entry from the extracellular milieu in mCRC cells. In agreement with this observation, removal of extracellular Ca^2+^ in plateauing cells caused the [Ca^2+^]_i_ to rapidly return to the baseline. Restoration of extracellular Ca^2+^ resumed the NaHS-evoked increase in [Ca^2+^]_i_ (Figure 4D). The molecular nature of the membrane pathway mediating this Ca^2+^ inflow was then investigated by a pharmacological approach.

### 2.3. TRPV1 Protein is Expressed and Mediates Extracellular Ca^2+^ Entry in Primary Cultures of mCRC Cells

H_2_S may induce extracellular Ca^2+^ entry by activating TRPV1 in a growing number of cell types [21,23,24,25]. Herein, we first sought to assess TRPV1 expression by exposing mCRC cells to capsaicin, a selective TRPV1 agonist [37,39,43,50,51], in presence of extracellular Ca^2+^. Accordingly, capsaicin induced a significant elevation in [Ca^2+^]_i_ that exhibited different temporal patterns (Figure 5), as previously described for NaHS (Figure 3A,B). The most frequent pattern observed consisted in a rapid increase in [Ca^2+^]_i_ which rapidly decayed to the baseline in the continuous presence of the agonist (blue trace in Figure 5A). This transient increase in [Ca^2+^]_i_ was detected in ≈75% of the cells (Figure 5B). In the remaining 25% (Figure 5B), the initial Ca^2+^ peak rapidly decayed to a sustained plateau phase that was maintained as long as capsaicin was present in the bath (green trace in Figure 5A). 

As expected, capsaicin (10 μM) did not evoke endogenous Ca^2+^ release in the absence of extracellular Ca^2+^ (0Ca^2+^), while the Ca^2+^ response resumed upon Ca^2+^ restitution to the perfusate (Figure 6A–D). Furthermore, capsaicin-induced extracellular Ca^2+^ entry was suppressed by preincubating the cells with specific TRPV1 antagonists, capsazepine (10 μM, 20 min), and SB-366791 (10 μM, 20 min) [37,39,50,52] (Figure 7A–C). Similarly, the Ca^2+^ response to capsaicin was inhibited by the less specific TRPV1 antagonist, ruthenium red (RuR) (10 μM, 20 min) [39,53,54,55] (Figure 7A–C).

Western blot and immunofluorescence analysis confirmed the expression of TRPV1 in mCRC cell (Figure 8). Immunoblots revealed a major band of approximately 75 kDa (Figure 8A and Appendix A), as also detected in other cell types [56,57,58]. Immunocytochemical analysis, performed using a primary anti-TRPV1 antibody and a Tetramethylrhodamine-isothiocyanate (TRITC)-conjugated secondary antibody, confirmed the expression of TRPV1 in mCRC cells (Appendix A). In control samples, stained with secondary antibody in the absence of primary antibody, no significant signal was detected using the same instrument settings (Appendix A). Confocal microscopy analysis revealed that red fluorescently-labeled TRPV1 is partially localized at the same focal plane of the plasma membrane, here labeled with green fluorescent dye (Figure 8B). Altogether, these results demonstrate that TRPV1 is expressed and able to mediate extracellular Ca^2+^ entry in mCRC cells.

### 2.4. TRPV1 Mediates H_2_S-Induced Extracellular Ca^2+^ Entry in Primary Cultures of mCRC Cells

In order to assess whether TRPV1 mediates H_2_S-evoked extracellular Ca^2+^ entry, mCRC cells were pre-treated with capsazepine (10 µM, 20 min) or SB-366791 (10 μM, 20 min) and then challenged with NaHS (100 µM). As shown in Figure 9A, this treatment dramatically impaired NaHS-evoked extracellular Ca^2+^ entry, thereby confirming that TRPV1 mediates the Ca^2+^ response to H_2_S. Capsazepine and SB-366791 dramatically reduced the percentage of responding cells (Figure 9B) and the amplitude of the residual Ca^2+^ signal in the responding ones (Figure 9C). In agreement with this observation, NaHS-evoked extracellular Ca^2+^ entry was also erased by the less specific TRPV1 inhibitor, ruthenium red (10 µM, 30 min) (Figure 9). 

Subsequently, we silenced TRPV1 expression by using a selective small interfering RNA (siTRPV1) (Appendix A). The efficacy of TRPV1 deletion in mCRC cells was confirmed by evaluating TRPV1 protein expression as compared to mCRC cells transfected with a scrambled construct (Figure 10). Preliminary experiments revealed that the Ca^2+^ response to capsaicin (10 µM) was significantly (*p* < 0.05) reduced in silenced cells (Figure 10A,B). Subsequently, we found that genetic deletion of TRPV1 remarkably reduced also NaHS-evoked extracellular Ca^2+^ entry (Figure 10C,D). These results indicate that TRPV1 channels play a crucial role in mediating H_2_S-induced extracellular Ca^2+^ influx in mCRC cells.

H_2_S has also been shown to activate VGCCs [21,29,30]. However, the Ca^2+^ response to NaHS (100 µM) was not affected either by nifedipine (10 µM, 30 min) or by Zn^2+^ (100 µM, 15 min) (Appendix A), which, respectively, block L- and T-type VGCCs [59,60]. A recent investigation showed that intracellular Ca^2+^ release from the endoplasmic reticulum (ER) in mCRC cells may fall below the resolution limit of epifluorescence Ca^2+^ imaging and lead to store-operated Ca^2+^ entry (SOCE) [36]. However, the pharmacological blockade of SOCE with the selective inhibitor, BTP-2 (20 µM, 30 min) [35,36], did not impair the increase in [Ca^2+^]_i_ evoked by NaHS (100 µM) (Appendix A). These findings are consistent with the notion that TRPV1 plays a major role in the onset of the Ca^2+^ response to H_2_S. 

In agreement with these findings, capsaicin-evoked Ca^2+^ signals were significantly (*p* < 0.05) larger in mCRC cells as compared to non-neoplastic an pCRC cells (Appendix A). Nevertheless, TRPV1 expression did not significantly differ between mCRC and non-neoplastic cells (Appendix A). 

### 2.5. The Reverse-Mode of NCX Contributes to H_2_S-Induced Extracellular Ca^2+^ Entry in Primary Cultures of mCRC Cells

Non-selective cation channels, such as TRPV1, which mediate also Na^+^ inrush into the cells, may elevate [Ca^2+^]_i_ by inverting the mode of operation of NCX [61,62]. Indeed, Na^+^ accumulation beneath the plasma membrane may locally switch the exchanger in the reverse direction (3Na^+^ out: 1 Ca^2+^ in). We, therefore, sought to assess whether NCX sustains TRPV1-mediated Ca^2+^ entry in mCRC cells [63] In order to inhibit NCX activity, extracellular Na^+^ was replaced by an equimolar amount of N-Methyl-D-glucamine (0Na^+^), that cannot substitute Na^+^ to sustain Ca^2+^ transport across the plasma membrane, as described elsewhere [62] This maneuver caused a transient increase in [Ca^2+^]_i_ due to the reversal of Na^+^ gradient and reversal of NCX activity into the reverse (i.e., Ca^2+^ entry) mode (Figure 11A). At the end of the Ca^2+^ transient, NCX could no longer mediate extracellular Ca^2+^ entry as Na^+^ was no longer present in the extracellular solution. Under this condition, the Ca^2+^ response to NaHS was abolished in 179 out of 245 cells (73%) (Figure 11A,B), whilst most cells were activated by the H_2_S donor upon restitution of extracellular Na^+^ (Figure 11A). The amplitude of the Ca^2+^ signal in the responding cells (66 out of 245; 27%) was significantly (*p* < 0.05) smaller as compared to the control (Figure 11C). In order to confirm NCX role in NaHS-dependent Ca^2+^ signaling, we probed the effect of KB-R 7943 (20 μM, 20 min), a selective inhibitor of the reverse-mode NCX at this concentration [62,64]. In agreement with the results presented above, NaHS failed to elicit an [Ca^2+^]_i_ elevation in 146 out of 182 (80.2%) cells pre-treated with the blocker (Figure 11D,E). In addition, the amplitude of the Ca^2+^ response was significantly lower in the fraction of cells activated by the H_2_S donor in presence of KB-R 7943 (Figure 11F). 

Furthermore, removal of extracellular Na^+^ (0Na^+^) and KB-R 7943 (20 μM, 20 min) interfered with the Ca^2+^ response to direct TRPV1 activation with capsaicin (10 μM). Under 0Na^+^ conditions, capsaicin induced a discernible increase in [Ca^2+^]_i_ in 81 out of 144 cells (Figure 12A–C), whereas the magnitude of the Ca^2+^ signal in responding cells was significantly (*p* < 0.05) reduced (Figure 12D). Likewise, capsaicin-induced intracellular Ca^2+^ signals were absent in 53 out of 134 cells pretreated with KB-R 7943 (Figure 12E,F), whereas the magnitude of the Ca^2+^ signal in responding cells was also significantly (*p* < 0.05) reduced (Figure 12G). Taken together, these findings hint at NCX as a key mediator of NaHS-elicited Ca^2+^ inflow in mCRC cells upon TRPV1 activation.

### 2.6. Exogenous H_2_S Suppresses Proliferation, but Does Not Induce Apoptosis, in mCRC Cells through TRPV1 Activation

Emerging evidence suggested an inhibitory effect of exogenous application of H_2_S in many cancer cell lines and, in particular, in CRC cell lines [16,17,18,19]. To study the effect of NaHS on mCRC proliferation, we sought to exploit the Trypan blue exclusion assay. Figure 13 showed the effect on mCRC viability of increasing NaHS concentrations (50, 100, and 200 µM). We found a significant (*p* < 0.05) reduction in mCRC number after 24 h preincubation in the presence of 100 µM and 200 µM NaHS, while 50 µM NaHS did not affect mCRC cell viability (Figure 13A). The proliferation rate underwent a further slight reduction after 72 h preincubation in the presence of 200 µM NaHS (Figure 13A). In order to assess the role of extracellular Ca^2+^ entry, we assessed the anti-proliferative effect of NaHS in the presence of capsazepine (10 µM), to inhibit TRPV1, and KB-R 7943 (10 µM), to block the reverse mode NCX. As depicted in Figure 13B, neither capsazepine (10 µM, 20 min) nor KB-R 7943 (10 µM, 30 min) alone affected proliferation in mCRC cells as compared to control cells. However, both drugs prevented the inhibitory effect of exogenous NaHS on mCRC cell viability at 100 µM. These results strongly indicate that exogenous administration of H_2_S could reduce proliferation in mCRC cells by triggering Ca^2+^ influx through activation of TRPV1 followed by recruitment of the reverse mode NCX.

A biparametric analysis with Annexin V and 7-amino-actinomycin D (7-AAD) was then carried to evaluate whether 100 µM NaHS inhibited mCRC cell proliferation by inducing apoptosis (Figure 13C), as shown in other tumor cell types [12,14,15,41]. Nevertheless, statistical analysis revealed that 72 h preincubation with 100 µM NaHS did not induce mCRC cell apoptosis (Figure 13D). Notably, a significant percentage of mCRC cells was found to be in late apoptosis under control conditions and exposure to 100 µM NaHS did not exacerbate the extent of mCRC cell death (Figure 13C,D). Depolarization of mitochondrial potential (ΔΨ_m_) is regarded an widespread early marker of apoptosis [65,66]. Consistent with previous findings, exposure to 100 µM NaHS for 72 h did not significantly change in ΔΨ_m_ mCRC cells loaded with tetramethylrhodamine, methyl ester (TMRM) (Figure 13E). Therefore, exogenous H_2_S did not cause mCRC cell death. 

### 2.7. Exogenous H_2_S Does Not Stimulate Phosphorylation Cascades in mCRC Cells

Finally, we evaluated whether the administration of exogenous H_2_S stimulate a number of intracellular signaling pathways which are known to control mCRC cell proliferation, migration and survival [34,68]. However, NaHS (100 µM) did not promote the phosphorylation of Akt, extracellular signal-regulated kinases 1 and 2 (Erk 1 and Erk 2, respectively) and of the mammalian target of rapamycin (mTOR) (Figure 14 and Appendix A). These data are consistent with the observation that H_2_S suppresses, rather than enhancing, mCRC cell proliferation.

## 3. Discussion

Exogenous delivery of H_2_S is emerging as an alternative strategy to treat multiple types of malignancies [12,14,15], including CRC [69]. The mechanism whereby exogenous H_2_S inhibits CRC proliferation is still unclear, although it could involve an increase in [Ca^2+^]_i_ [20]. Furthermore, the anticancer effect of H_2_S remains to be confirmed in patients-derived CRC cells, as recent studies revealed that intracellular Ca^2+^ signals may drive cell fate in multiple commercially available cancer cell lines, but not in primary cultures established from tumors as different as glioblastoma [70], renal cellular carcinoma [71], and CRC [36]. In addition, only a few studies described H_2_S-induced intracellular Ca^2+^ signals in cancer cells [20,49], but they failed to identify the main mediator(s) of the Ca^2+^ response. The present investigation provided the first evidence that H_2_S inhibits proliferation in primary cultures of mCRC cells by inducing extracellular Ca^2+^ entry through TRPV1. TRPV1-mediated Ca^2+^ entry is, in turn, sustained by the reverse mode NCX. This finding endorses the view that exogenous delivery of H_2_S represents a promising strategy to treat CRC.

### 3.1. H_2_S-Evoked Intracellular Ca^2+^ Signals in Primary Cultures of mCRC Cells

Exogenous administration of H_2_S in the form of the widely employed H_2_S-donor, NaHS, reliably increased the [Ca^2+^]_i_ in primary cultures of mCRC cells. Conversely, the Ca^2+^ responses to NaHS were remarkably smaller in non-neoplastic and pCRC cells. This preliminary investigation revealed that mCRC cells displayed a higher sensitivity of exogenous delivery of H_2_S. An increase in [Ca^2+^]_i_ was also induced by H_2_S release from another donor, GYY4137, in the CRC cell line, DLD1 [20]. While this investigation did not investigate the dose-response relationship and the kinetics of the Ca^2+^ response to H_2_S release [20], herein we first found that, while the Ca^2+^ signal arose at low NaHS concentration ([NaHS]) (i.e., 5 µM), the percentage of responding cells and the magnitude of the initial Ca^2+^ peak decreased by further reducing the [NaHS] to 25–50 µM. Nevertheless, the Ca^2+^ response was fully restored by increasing the [NaHS] to 75-300 µM. A similar U-shaped dose-response relationship has been described for NaHS-evoked intracellular Ca^2+^ signals in the native endothelium of rat aorta [62] and EA.hy926 cells, while a bell-shaped pattern has been reported in breast tumor-derived endothelial cells (B-TECs) [7]. The peculiar U-shaped pattern of the dose-response relationship that H_2_S exhibits in several cellular models has been attributed to the wide array of signaling pathways recruited by this gasotransmitter [72,73]. It should also be noticed that H_2_S may differently affect the Ca^2+^ handling machinery even within the same cell type [21,22]. For instance, NaHS stimulated inositol-1,4,5-trisphosphate (InsP_3_) receptors (InsP_3_Rs) in Ea.hy926 cells [7] and saphenous vein-derived endothelial cells [74], while it inhibited InsP_3_-dependent Ca^2+^ release in rat aortic endothelium [7] and was ineffective in human endothelial colony forming cells [7]. Furthermore, NaHS was found to both inhibit [75] and activate [76] voltage-gated Ca^2+^ entry in rat cardiac myocytes. Likewise, NaHS was able to facilitate [77] or block [78] CaV3.2 ectopically expressed in HEK-293 cells. Such a variable effect exerted by H_2_S could be due to the different cysteine residues that can be sulfhydrated within the same protein channel, thereby exerting distinct effects on its activity [22]. We, therefore, focused on 100 µM NaHS, as mCRC cells displayed the highest Ca^2+^ sensitivity to this [NaHS] and preliminary experiments showed that NaHS did not affect mCRC cell viability at lower doses.

One hundred µM NaHS evoked two main intracellular Ca^2+^ signatures in mCRC cells: a transient increase in [Ca^2+^]_i_, which occurred in ≈75% of the cells, and a biphasic Ca^2+^ signal, which arose in the remaining lower fraction of cells. Different patterns of intracellular Ca^2+^ signals were also elicited by NaHS in rat aortic endothelium [62] and endothelial cells harvested from human saphenous vein (SVECs) [74]. Unlike non-cancer cells [62,74], the Ca^2+^ response to NaHS was reversible and did not desensitize after the first stimulation, which suggests that the molecular trigger of the Ca^2+^ signal is different and/or that the recovery from H_2_S-induced modifications (most likely, sulfhydration) is faster in mCRC cells. As discussed in more detail in the next paragraph, our evidence hints at TRPV1 as the main responsible for the onset of the Ca^2+^ response to NaHS in mCRC cells.

### 3.2. Evidence that TRPV1 and Reverse Mode NCX Mediate H_2_S-Evoked Intracellular Ca^2+^ Signals in Primary Cultures of mCRC Cells

It has been demonstrated that H_2_S may increase the [Ca^2+^]_i_ by both mobilizing ER stored Ca^2+^ through InsP_3_Rs and ryanodine receptors (RyRs) and activating extracellular Ca^2+^ entry [7,21,22,74]. However, NaHS evoked failed to induce any discernible Ca^2+^ signal upon removal of extracellular Ca^2+^, while the Ca^2+^ response immediately resumed upon Ca^2+^ restitution to the bathing solution. While this finding does not rule out the possibility that a local Ca^2+^ signal is induced by NaHS, and is missed by our epifluorescence detection system, it does demonstrate that the bulk increase in [Ca^2+^]_i_ is triggered by extracellular Ca^2+^ entry. The same observation has been reported in another cancer-derived cell line, i.e., B-TECs [49], although the underlying signaling pathway has not been uncovered. 

#### 3.2.1. TRPV1 Triggers the Ca^2+^ Response to NaHS in Primary Cultures of mCRC Cells

The following pieces of evidence indicate that NaHS-evoked extracellular Ca^2+^ entry in primary cultures of mCRC cells is triggered by TRPV1. First, the Ca^2+^ response to NaHS was erased by blocking TRPV1 with the selective antagonists, capsazepine and SB-366791 [37,39,50,52], and the less selective blocker ruthenium red [39,40]. Moreover, NaHS-evoked extracellular Ca^2+^ entry was significantly reduced by the genetic deletion of TRPV1 by a selective siTRPV1. Second, TRPV1 protein was abundantly expressed in mCRC cells, as demonstrated by immunoblotting and confocal microscopy analysis of TRPV1 expression. Likewise, TRPV1 transcript has been reported in the CRC cell line [79], HT29, and TRPV1 protein has been detected in CRC tissue [42]. The theoretical molecular weight (MW) of TRPV1 protein is 90 kDa [37]. Our TRPV1 antibody revealed a protein species of 75 kDa, as previously reported in rodent urothelium [58], retinal ganglion cells [57] and astrocytes [56], normal human bronchial epithelial cells [80], mouse and chicken chondrogenic cells [81]. As discussed in [81], multiple TRPV1 splice variants were detected [82,83,84,85,86]. Furthermore, TRPV1 proteins presents several sites for post-translational modifications, which induce alterations in the MW of the channel protein [86,87,88,89]. Third, the dietary agonist capsaicin, which is widely exploited to monitor TRPV1 activation [37,39,43,50,51], induced an increase in [Ca^2+^]_i_ that was also sensitive to capsazepine, SB-366791 and ruthenium red, and to genetic deletion of TRPV1. Notably, capsaicin evoked two distinct patterns of intracellular Ca^2+^ signals that strongly resembled those induced by NaHS: a transient Ca^2+^ elevation in ≈75% cells and a biphasic Ca^2+^ response in the remaining ≈25% cells. One to ten µM capsaicin was shown to elicit both transient [52,90] and long-lasting [51,91,92,93,94] intracellular Ca^2+^ signals upon TRPV1 activation. The variable kinetics of the Ca^2+^ response to capsaicin (and NaHS) is likely to depend on the different extent of TRPV1 inactivation. As widely discussed in [40], extracellular Ca^2+^ entry could inactivate the channel by inducing TRPV1 dephosphorylation [95], depleting phosphatidylinositol 4,5-bisphosphate levels [96], or calmodulin binding [97]. It is, therefore, conceivable that the Ca^2+^-dependent mechanisms of TRPV1 inactivation are uncoupled from extracellular Ca^2+^ entry in the majority of mCRC cells. In addition, it has been suggested that increasing the strength of capsaicin stimulation results in TRPV1 pore dilation, thereby increasing single channel conductance and prolonging the duration of the ensuing increase in [Ca^2+^]_i_ [93,98]. However, this mechanism is unlikely to explain the heterogeneity in the Ca^2+^ response to capsaicin reported in the present investigation as this was observed by presenting mCRC cells with the same agonist concentration. Ten µM capsaicin was found to stimulate TRPV1 and induce long-lasting elevations in in MCF-7 breast cancer cells [43], while the effect of TRPV1 stimulation on prostate cancer is more controversial. Indeed, 50 µM capsaicin induced intracellular Ca^2+^ oscillations in some prostate cancer cell lines (DU 145 and PC-3) in one study [50], while it failed to increase the [Ca^2+^]_I_ in another one [51]. Capsaicin increased resting also in CRC HCT116 cells [42], but this study did not evaluate the kinetics and underlying mechanisms of the Ca^2+^ signal. 

#### 3.2.2. The Reverse Mode NCX Sustains the Ca^2+^ Response to NaHS in Primary Cultures of mCRC Cells

The following pieces of evidence indicate that the reverse (i.e., Ca^2+^ entry) mode NCX sustains NaHS- and capsaicin-induced extracellular Ca^2+^ entry. First, replacement of extracellular Na^+^ with an equimolar amount of choline induced a transient increase in [Ca^2+^]_i_, which reflects the directional switch of NCX from the forward (i.e., Ca^2+^ exit) to the reverse (i.e., Ca^2+^ entry) mode upon reversal of Na^+^ gradient across the plasma membrane [62,99]. Second, only a modest fraction (≈25%) of mCRC cells displayed a Ca^2+^ signal in response to NaHS and capsaicin under 0Na^+^ conditions. The amplitude of the Ca^2+^ elevation was, however, significantly lower whereas its duration was dramatically curtailed. Third, the selective blockade of the reverse mode NCX with KB-R mimicked the effect of Na^+^ withdrawal on NaHS- and capsaicin-induced intracellular Ca^2+^ signals [62,64], thereby confirming that the reverse mode NCX is recruited by TRPV1-mediated extracellular Na^+^ entry to sustain the ensuing increase in [Ca^2+^]_i_. These data are consistent with those reported in native endothelium of rat aorta [62] and confirm a recent report on in the CRC cell line, DLD1 [20]. In these cells, exogenous delivery of H_2_S through GYY4137 induced an increase in resting [Ca^2+^]_i_ that was associated to the switch of NCX activity into the reverse mode. Furthermore, earlier work demonstrated that H_2_S release was able to increase the expression levels of NCX1 [100], the main NCX isoform expressed in CRC cells [20]. It is, therefore, possible to conclude that TRPV1 and NCX may by physically coupled in primary mCRC cells, as also described for TRP Melastatin 4 [101] and TRP Canonical 6 [102]. This feature might contribute to explaining why capsaicin- and NaHS-evoked extracellular Ca^2+^ entry is larger in mCRC cells although TRPV1 protein expression was similar in mCRC and non-neoplastic cells. We speculate that TRPV1 channels may be uncoupled from NCX (possibly due to lower NCX expression) in non-neoplastic and pCRC cells, thereby attenuating the amplitude of the Ca^2+^ response to TRPV1 activation. Experiments in our laboratories are under way to assess this important issue.

### 3.3. H_2_S Inhibits Proliferation in Primary Cultures of mCRC Cells

Exogenous delivery of H_2_S has been shown to suppress proliferation in several types of cancer cell lines, including hepatocellular carcinoma (Hep G2), human cervical carcinoma (HeLa), breast adenocarcinoma (MCF-7), osteosarcoma (U2OS), human thyroid carcinoma (Nthy-ori3-1), and erythroleukemia (K562) [19,41,103]. In addition, exogenously added H_2_S impaired cell viability in a wide number of CRC cell lines, including HT-29, Caco-2, HCT-116, SW1116, HCT116, and DLD1 [16,17,18,19]. Herein, we confirmed that exogenous H_2_S significantly reduced proliferation in primary cultures of mCRC cells exposed to 100–200 µM NaHS for 24 h. Of note, there was not further decrease in mCRC cell survival when the exposure to NaHS was prolonged to 72 h, which suggests the activation of protective signaling pathways. For instance, 24–48 h treatment with 400–1000 µM NaHS induced protective autophagy by inhibiting the mammalian target of rapamycin, thereby hampering the concurrent anti-mitogenic effect on CRC cell lines [17]. The anti-proliferative effect of NaHS was rescued by preventing TRPV1-induced extracellular Ca^2+^ entry wither with capsazepine or KB-R 7943, as recently reported in leukemia, breast cancer, cervical carcinoma cell lines [41]. TRPV1 has been shown to inhibit cancer cell proliferation by promoting apoptosis [42]. However, 72 h treatment with NaHS did not increase the basal rate of mCRC cell apoptosis, neither promoted mitochondrial depolarization, which represents an established marker of apoptosis. Conversely, TRPV1-mediated Ca^2+^ entry recruited calcineurin to activate p53 and induce apoptosis in the CRC HCT116 cell line [42]. As discussed elsewhere [35,104], the discrepancy between this report and our findings is consistent with the emerging notion that the same components of the Ca^2+^ toolkit could induce distinct outcomes in primary cultures of mCRC cells [36] and commercially available CRC cell lines [31,105]. Furthermore, the Ca^2+^ response evoked by exogenous administration of H_2_S displayed transient kinetics in the majority of mCRC cells. Conversely, pro-apoptotic Ca^2+^ signals usually consist in prolonged elevations in [Ca^2+^]_i_, which persist as long as the stimulus is presented to the cells [66,106,107,108,109], as we have recently shown in cisplatin-treated glioblastoma cells [65]. 

Nevertheless, data obtained from both primary cultures and immortalized cell lines support the anticancer effect of exogenous H_2_S delivery in CRC. Interestingly, a recent report showed that, in bladder cancer cell lines, TRPV1-mediated extracellular Ca^2+^ entry prevented the nuclear translocation of proliferating cell nuclear antigen (PCNA), a 29 kDa protein which serves as auxiliary protein for accessory protein for DNA polymerase δ (Polδ) and DNA polymerase ε (Polε) [110,111]. Future work will have to assess whether this signaling pathway is recruited by exogenous H_2_S to inhibit proliferation also in mCRC cells. Currently, we provided the evidence that the administration of exogenous H_2_S did not activate the Akt, Erk 1/2, and mTOR signaling pathways. 

## 4. Materials and Methods 

### 4.1. Expansion of Tumor Cells 

Patients (>18 years) suffering from mCRC who had undergone surgery intervention to dissect primary tumor and/or liver metastases were enrolled. All the patients signed an informed consent. Tumor specimens were processed as previously described [36,112]. The present investigation has been approved by the Ethical Committee of the IRCCS Foundation Policlinico San Matteo (protocol number: 20190069408). Briefly, tumor samples were treated with Tumor dissociation Kit (Miltenyi BIOTEC, Bergisch Gladbach, Germany) and then disaggregated with the gentleMACS Dissociator (Miltenyi BIOTEC, Bergisch Gladbach, Germany) according to the manufacturers’ instructions. Tumor cells recovered from filtration for removing were resuspended at a concentration of 0.5–1 × 10^6^ cells/mL of CellGro SCGM (CellGenix, Freiburg, Germany), supplemented with 20% FBS, 2 mM L-glutamine (complete medium) (Life Technologies, Inc. Monza, Italy) and cultured in 25 cm^2^ tissue flasks (Corning, Stone, England) at 37 °C and 5% CO_2_. The culture medium was changed twice a week and cellular homogeneity evaluated microscopically every 24–48 h, when cultures reaching about 75–100% confluence were subjected to trypsinization with 0.25% trypsin and 0.02% EDTA (Life Technologies, Inc.) in a calcium/magnesium-free balanced solution. The culture medium was changed twice a week and cellular homogeneity was confirmed by using a light microscope every 24–48 h. In particular, we cryopreserved several vials at passage T0 (after disaggregation) and at passage T1. These vials were then further thawed for further experiments. Using this approach, the experiments were carried out with tumor cells that underwent no more than 3–4 in vitro passages, to avoid problems related to long-term culture of cancer cells. Cells were cryopreserved in 90% FBS and 10% dimethyl sulfoxide and stored in liquid nitrogen for further experiments. To confirm the neoplastic origin of cultured cells passaged 3–5 times, we carried out morphological and immunocytochemical analysis, as described in [112,113]. Non-neoplastic cells were derived from tissue a sample of healthy colon tissue near the tumor. Cells were expanded and their non-neoplastic nature was assessed by morphological and immunocytochemical analysis. 

### 4.2. Solutions for Intracellular Ca^2+^ Recordings

The composition of the physiological salt solution (PSS) was the following (in mM): 150 NaCl, 6 KCl, 1.5 CaCl_2_, 1 MgCl_2_, 10 Glucose, 10 HEPES. In Ca^2+^-free solution (0Ca^2+^), Ca^2+^ was replaced with 2 mM NaCl, and 0.5 mM EGTA was added. Solutions were titrated to pH 7.4 with NaOH. The osmolality of PSS was measured with an osmometer (WESCOR 5500, Logan, UT, USA) and was equal to 338 mmol/kg.

### 4.3. [Ca^2+^]_i_ Measurements

Ca^2+^ imaging in mCRC cells was carried out as described elsewhere [34,36]. Briefly, mCRC cells were loaded with 4 µM fura-2 acetoxymethyl ester (Fura-2/AM; 1 mM stock in dimethyl sulfoxide) in PSS for 30 min at 37 °C and 5% CO_2_. The cells were maintained in the presence of Fura-2 for 30 min at 37 °C and 5% CO_2_ saturated humidity. After de-esterification in PSS for 15 min, the coverslip (8 mm) was mounted on the bottom of a Petri dish and the cells observed by an upright epifluorescence Axiolab microscope (Carl Zeiss, Oberkochen, Germany) equipped with a Zeiss ×40 Achroplan objective (water-immersion, 2.0 mm working distance, 0.9 numerical aperture). The cells were alternately excited at 340 nM and 380 nm by using a filter wheel (Lambda 10, Sutter Instrument, Novato, CA, USA). The emitted fluorescence was detected at 510 nm by using an Extended-ISIS CCD camera (Photonic Science, Millham, UK). Custom software, working in the LINUX environment, was used to drive the camera and the filter wheel, and to measure and plot on-line the fluorescence from 10 up to 40 rectangular “regions of interest” (ROI), each corresponding to a well-defined single cell. The [Ca^2+^]_i_ was monitored by measuring, for each ROI, the ratio of the mean fluorescence emitted at 510 nm when exciting alternatively at 340 and 380 nm (Ratio (F_340_/F_380_)). An increase in [Ca^2+^]_i_ causes an increase in the ratio [36]. Ratio measurements were performed and plotted on-line every 3 s. The experiments were performed at room temperature (22 °C).

### 4.4. SDS-PAGE and Immunoblotting

Cells were lysed in ice-cold RIPA buffer (50 mM TRIS/HCl, pH 7.4, 150 mM NaCl, 1% Nonidet P40, 1 mM EDTA, 0.25% sodium deoxycholate, 0.1% SDS) added of protease and phosphatase inhibitors. Upon protein quantification, the samples were dissociated by addition of half volume of SDS-sample buffer 3× (37.5 mM TRIS, pH 8.3, 288 mM glycine, 6% SDS, 1.5% DTT, 30% glycerol, and 0.03% bromophenol blue), separated by SDS-PAGE on a 10% or 6% polyacrylamide gel, and blotted on a PVDF membrane. Membrane probing was performed using the different antibodies diluted 1:1000 in TBS (20 mM Tris, 500 mM NaCl, pH 7.5) containing 5% BSA and 0.1% Tween-20 in combination with the appropriate horseradish peroxidase (HRP)-conjugated secondary antibodies (1:2000 in Phosphate Buffered Saline (PBS) plus 0.1% Tween-20). The following antibodies were used: anti-TRPV1 (ab3487) from ABCAM (Cambridge, UK), HRP-conjugated anti-GAPDH (sc365062), anti-Akt 1/2/3 (sc8312), anti-Erk 2 (sc-154) and anti-mTOR (sc1549) from Santa Cruz Biotechnology (Dallas, TX, USA); the specific phospho-antibodies: phospho-Akt (Ser473) (#9271), phospho-Erk 1/2 (Thr 202/Tyr 204) (#9101) and phosphor-mTOR (Ser 2481) (#2974) from Cell Signaling Technology (Danvers, MA, USA). The chemiluminescence reaction was performed using Immobilon Western (Millipore, Burlington, MA, USA) and images were acquired by ChemiDoc XRS (Bio-Rad, Hercules, CA, USA). Membrane was then re-probed with an HRP-conjugated anti-GAPDH antibody as equal loading control. 

### 4.5. Confocal Microscopy and Immunofluorescence

Cells were grown for 24 h on 15 mm glass coverslips in a 12-well plate. Samples were then fixed with 3% PFA in PBS, permeabilized with 0.25% ice-cold TRITON X100 in PBS and blocked with 1% BSA in PBS. The coverslips were stained with a rabbit anti-human TRPV1 antibody (1:250, ab3487-ABCAM; Cambridge, UK) for 1 h at room temperature and then incubated with TRITC-anti rabbit secondary antibody (1:500, ab6718-ABCAM; Cambride, UK) in the dark for 1 h at room temperature. Negative controls were prepared by omitting the primary antibody. Nuclei were stained with 1 µg/mL Hoechst 33342 (#4082-Cell Signaling Technologies; Danvers, MA, USA) and cell membrane with 2 µM PKH67 (MINI67–Sigma-Aldrich; Milan, Italy). Finally, the coverslips were mounted on glass slides using ProLong Gold antifade reagent (Invitrogen, Carlsbad, CA, USA) as mounting medium. Immunofluorescence images were acquired using an Olympus BX51 microscope, whereas the confocal microscopy images were captured using a Leica TCS SP8 Microscope and analyzed with LAS X software (Leica Microsystems GmbH, Wetzlar, Germany).

### 4.6. Gene Silencing

The siRNA targeting TRPV1 was purchased from Sigma-Aldrich Inc. MISSION esiRNA (human TRPV1) (EHU073721). Scrambled siRNA was used as negative control. In brief, once the seeded mCRC cells had reached 50% confluency, the medium was replaced with Opti-MEM, the serum in the medium reduced without antibiotics (Life Technologies, Milan, Italy). siRNAs (100 nM final concentration) were diluted with Opti-MEM I reduced serum medium and mixed with Lipofectamine™ RNAiMAX transfection reagent (Life Technologies, Milan, Italy) pre-diluted in Opti-MEM), according to the manufacturer’s instructions. After 20 min incubation at room temperature, the mixtures were added to the cells and incubated at 37 °C for 5 h. Transfection mixes were then completely removed and fresh culture media was added again and silenced cells were used 48 h after transfection. The effectiveness of silencing was determined by immunoblotting (see Appendix A). 

### 4.7. Identification of Apoptotic Cells

Metastatic CRC cells at confluence were detached by trypsinization and incubated with FITC-conjugated Annexin V (5 µL/5 × 10^5^ cells) (Annexin V apoptosis Detection Kit, Invitrogen, Carlsbad, California, USA) and with 7-ADD (5 µL/5 × 10^5^ cells) (BD Pharmingen, San Diego, CA, USA). Metastatic CRC cells maintained in the absence and presence of NaHS (100 µM, 72 h) were incubated for 15 min at resting temperature and then analyzed by Beckman Coulter Navios according to manufactures’ instructions.

### 4.8. Measurement of Mitochondrial Membrane Potential (ΔΨ_m_)

ΔΨ_m_ was evaluated with TMRM at a loading concentration sufficient to cause dye aggregation within the mitochondrial matrix and by using the same fluorescence detection system used to measure changes in Fura-2 fluorescence. The mCRC cells were loaded with 25 nM TMRM and 200 nM of cyclosporine H, in PSS for 30 min at 37 °C and 5% CO_2_. After washing in PSS, the coverslip was fixed to the bottom of a Petri dish and the cells were excited 480 nm, while the emitted light was detected at 510 nm. TRITC filter for live imaging has been utilized to examine the TMRM red-orange fluorescence, and round diaphragm was employed to augment the contrast. Measurements were carried out and plotted on-line every 10 s. All the recordings were carried out at 22 °C. 

### 4.9. Statistics

All the data have been obtained from mCRC cells plated on at least three coverslips and deriving from three distinct donors. Each trace shown is representative of multiple cells displaying a similar Ca^2+^ activity and deriving from at least three distinct donors. The peak amplitude of NaHs- and capsaicin-evoked Ca^2+^ signals was measured by evaluating the difference between the F_340_/F_380_ ratio at the peak of the Ca^2+^ response, and the mean F_340_/F_380_ ratio of 1 min baseline recording before agonist addition. Pooled data are presented as mean ± SE, and statistical significance (*p* < 0.05) was evaluated by the Student’s *t*-test for unpaired observations or one-way ANOVA analysis followed by the post-hoc Dunnett’s or Bonferroni tests, as required. The number of cells measured for each experimental condition is indicated in, or above, the corresponding bar histogram.

## 5. Conclusions

This study demonstrates for the first time that the H_2_S-releasing compounds NaHS elicit intracellular Ca^2+^ signals in primary cultures of mCRC, but not of non-neoplastic and pCRC cells, by stimulating TRPV1 to mediate extracellular Ca^2+^ entry and induce NCX to switch into the reverse (Ca^2+^ entry) mode. NaHS-induced TRPV1 activation, in turn, inhibited mCRC cell proliferation. The anti-proliferative effect of NaHS was not due to mCRC cell apoptosis. These data lend further support to the exogenous delivery of H_2_S as a novel therapeutic strategy to treat mCRC.

## Figures and Tables

**Figure 1 cancers-12-03338-f001:**
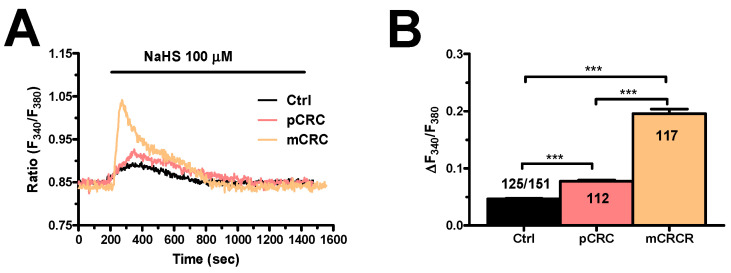
NaHS evokes intracellular Ca^2+^ signals in colorectal cancer (CRC) and non-neoplastic cells. (**A**), NaHS (100 µM) evoked intracellular Ca^2+^ signals in non-neoplastic (Control, Ctrl), primary CRC (pCRC) and metastatic CRC (mCRC) cells. (**B**), mean ± SE of the amplitude of the peak Ca^2+^ response induced by NaHS in the different cell types. One-way A analysis followed by the post-hoc Bonferroni test was used for Statistical comparison. In Panels B: *** *p* ≤ 0.001.

**Figure 2 cancers-12-03338-f002:**
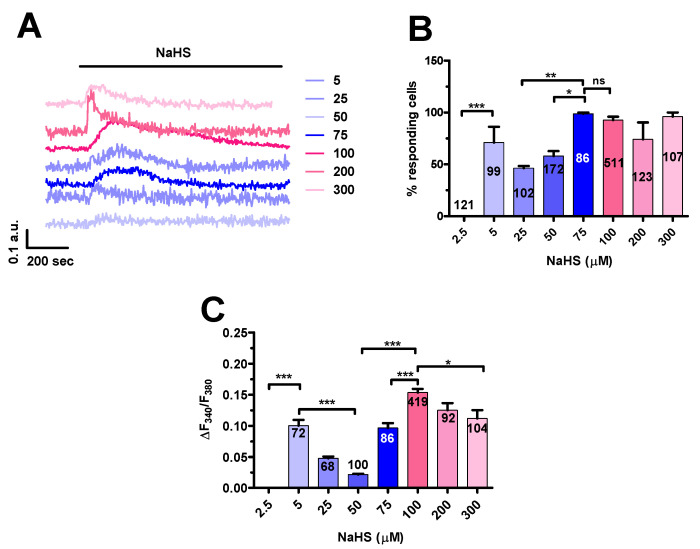
Dose-dependent effect of NaHS on [Ca^2+^]_i_ in mCRC cells. (**A**), intracellular Ca^2+^ signals evoked by increasing concentrations of NaHS in mCRC cells. Each dose-response relationship was carried out on cells from the same batch in three separate experiments. (**B**), mean ± SE of the percentage of cells presenting a discernible increase in [Ca^2+^]_i_ in the presence of different concentrations of NaHS. (**C**), mean ± SE of the amplitude of the peak Ca^2+^ response to different concentration of NaHS. One-way ANOVA analysis followed by the post-hoc Bonferroni test was used for Statistical comparison. In Panels B and C: *** *p* ≤ 0.001; ** *p* ≤ 0.01; * *p* ≤ 0.05; ns: not significant.

**Figure 3 cancers-12-03338-f003:**
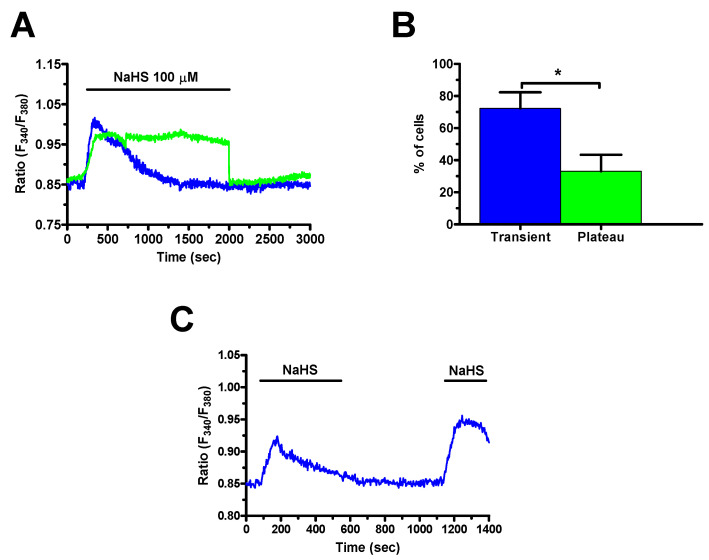
Distinct Ca^2+^ signals induced by NaHS in mCRC cells. (**A**), administration of 100 µM NaHS in the presence of extracellular Ca^2+^ triggered heterogeneous Ca^2+^ signals in mCRC cells, such as transient and long-lasting increases in [Ca^2+^]_i_. (**B**), mean ± SE of the percentage of cells presenting transient and long-lasting Ca^2+^ signals in 287 mCRC cells in response to 100 µM NaHS. (**C**), repetitive applications of NaHS (100 μM) resulted in Ca^2+^ signals similar amplitude and kinetics. Student’s *t*-test has been used for statistical comparison. The asterisk indicates *p* < 0.05.

**Figure 4 cancers-12-03338-f004:**
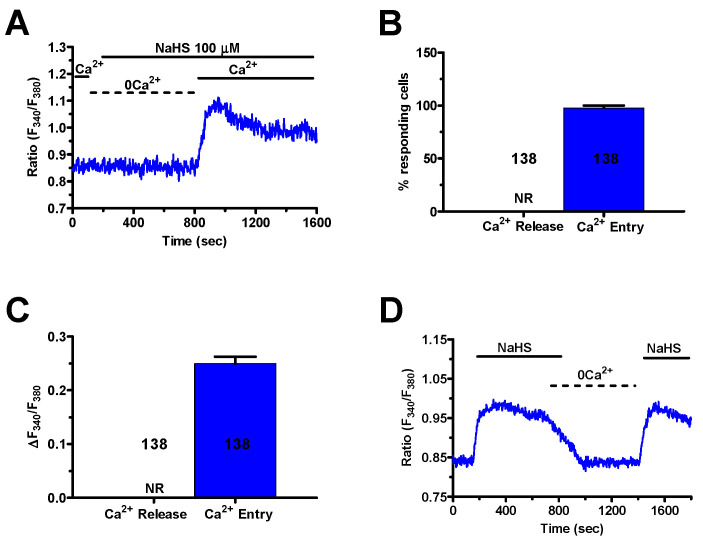
Extracellular Ca^2+^ entry mediates the Ca^2+^ response to NaHS. (**A**), NaHS (100 μM) failed to elicit intracellular Ca^2+^ levels when the cells were bathed in the absence of extracellular Ca^2+^ (0Ca^2+^), the [Ca^2+^]_i_ raised following re-addition of extracellular Ca^2+^. (**B**), mean ± SE of the percentage of mCRC cells presenting a discernible Ca^2+^ release or Ca^2+^ entry in response to 100 μM NaHS. (**C**), mean ± SE of the amplitude of Ca^2+^ release and Ca^2+^ entry induced by NaHS in mCRC cells. (**D**), removal of extracellular Ca^2+^ (0Ca^2+^) during the plateau phase caused the [Ca^2+^]_i_ to undergo a reversible decline to pre-stimulation levels.

**Figure 5 cancers-12-03338-f005:**
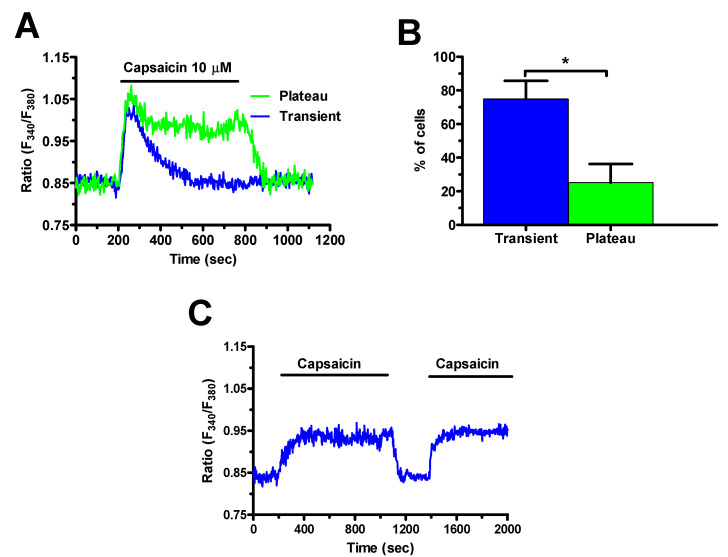
Capsaicin induces extracellular Ca^2+^ influx in mCRC cells. (**A**), administration of capsaicin (10 µM), a specific TRPV1 agonist, in the presence of extracellular Ca^2+^ induced both transient and long-lasting intracellular Ca^2+^ signals. (**B**), mean ± SE of the percentage of cells presenting transient and long-lasting Ca^2+^ signals in 170 mCRC cells in response to 10 µM capsaicin. Student’s *t*-test has been used for statistical comparison. The asterisk indicates *p* < 0.05. (**C**), repetitive applications of capsaicin (10 μM) resulted in Ca^2+^ signals with similar amplitude and kinetics.

**Figure 6 cancers-12-03338-f006:**
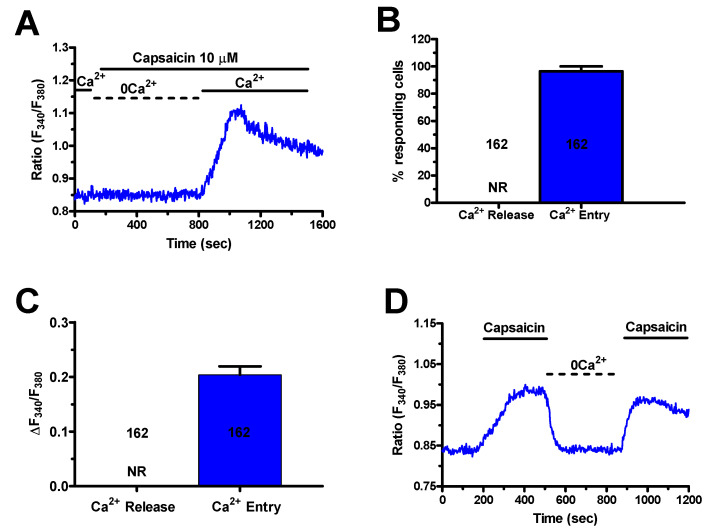
Extracellular Ca^2+^ entry mediates the Ca^2+^ response to capsaicin in mCRC cells. (**A**), application of capsaicin (10 µM) in the absence of extracellular Ca^2+^ (0Ca^2+^) did not elicit any detectable increase in [Ca^2+^]_i_, while the Ca^2+^ signal was recorded upon Ca^2+^ restitution to the bath. (**B**), mean ± SE of the percentage of mCRC cells presenting a discernible Ca^2+^ release or Ca^2+^ entry in response to 10 μM capsaicin. (**C**), mean ± SE of the amplitude of Ca^2+^ release and Ca^2+^ entry induced by capsaicin in mCRC cells. (**D**), removal of extracellular Ca^2+^ (0Ca^2+^) during the plateau phase caused the [Ca^2+^]_i_ to undergo a reversible decline to pre-stimulation levels.

**Figure 7 cancers-12-03338-f007:**
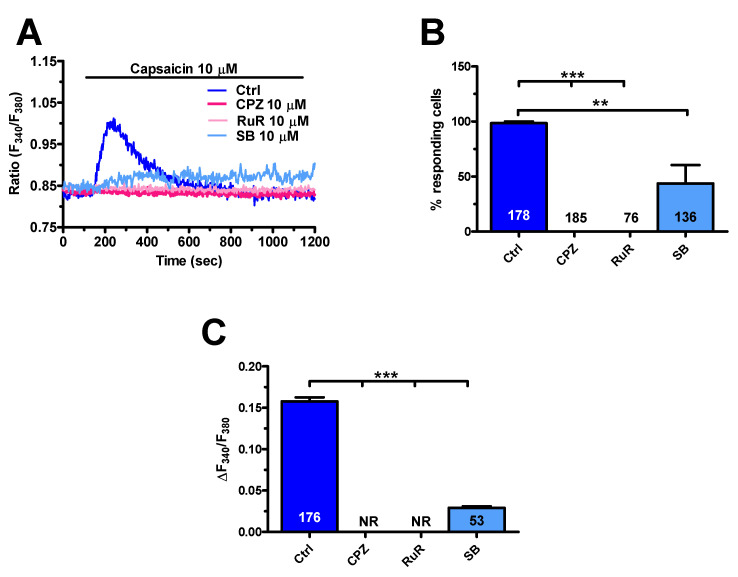
Pharmacological blockade of Transient Receptor Potential Vanilloid 1 (TRPV1) inhibited capsaicin-induced Ca^2+^ entry in mCRC. (**A**), transient Ca^2+^ signals induced by 10 µM of capsaicin were abrogated upon preincubation with the specific TRPV1 antagonists: capsazepine (CPZ; 10 µM, 20 min) and SB 366791 (SB; 10 µM, 20 min) and with the pan-specific TRPV antagonist ruthenium red (RuR; 10 µM, 20 min). (**B**), mean ± SE of the percentage of cells presenting a discernible Ca^2+^ response to capsaicin in the absence (Ctrl) and presence of TRPV1 blockers. (**C**), mean ± SE the amplitude of the Ca^2+^ response to capsaicin in the absence (Ctrl) and presence of TRPV1 blockers. One-way ANOVA analysis followed by the post-hoc Dunnett’s test was used for Statistical comparison. In Panels B and C: *** *p* ≤ 0.001; ** *p* ≤ 0.01.

**Figure 8 cancers-12-03338-f008:**
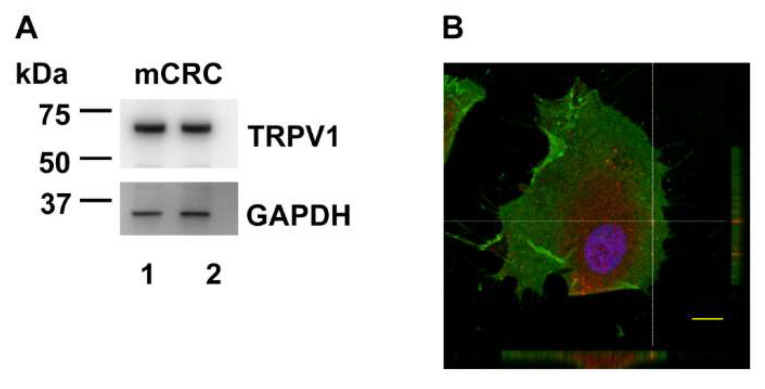
Expression and subcellular distribution of TRPV1 in mCRC cells. (**A**), immunoblotting analysis of TRPV1 expression in two samples of mCRC cells. Total lysates (15 µg) were separated by SDS-PAGE, blotted on polyvinylidene fluoride (PVDF) membrane and stained with anti TRPV1 antibody. Subsequent reprobing with anti-GAPDH antibody was performed as an equal loading control. (**B**), Confocal microscopy analysis of the TRPV1 distribution (red signal). Cell nuclei were stained with Hoechst 33342 (blue) and plasma membrane was labeled with PKH67 (green). Representative confocal middle z-section and orthogonal views are reported. Scale bar: 10 μm.

**Figure 9 cancers-12-03338-f009:**
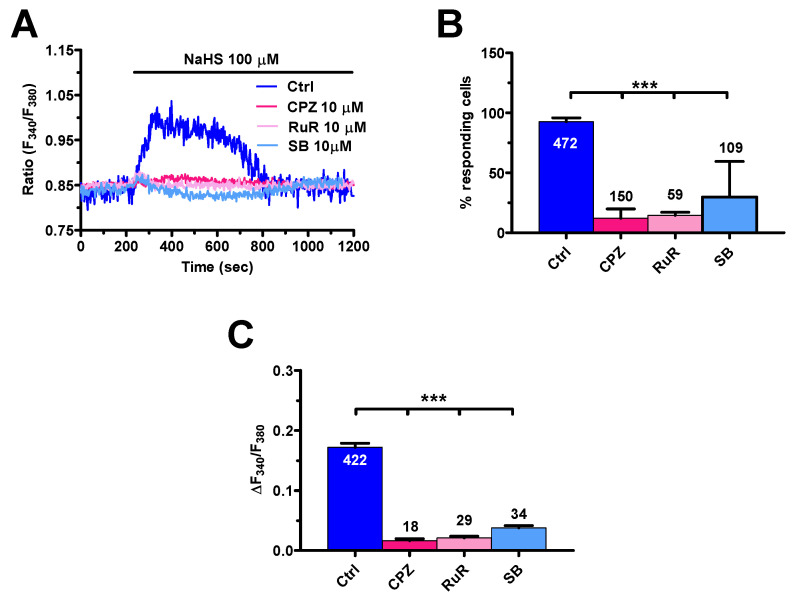
NaHS induces Ca^2+^ influx by activating TRPV1 in mCRC cells. (**A**), NaHS (100 µM) evoked an increase in [Ca^2+^]_i_ in the absence (Ctrl), but not in the presence, of the specific TRPV1 antagonists: capsazepine (CPZ; 10 µM, 20 min) and SB 366791 (SB; 10 µM, 20 min) and of the less specific TRPV1 blocker ruthenium red (RuR; 10 µM, 20 min). (**B**), mean ± SE of the percentage of cells presenting a discernible Ca^2+^ response to NaHS under the designated treatments. (**C**), mean ± SE the amplitude of NaHS-evoked Ca^2+^ influx under the designated treatments. One-way ANOVA analysis followed by the post-hoc Dunnett’s test was used for Statistical comparison. In Panels B and C: *** *p* ≤ 0.001.

**Figure 10 cancers-12-03338-f010:**
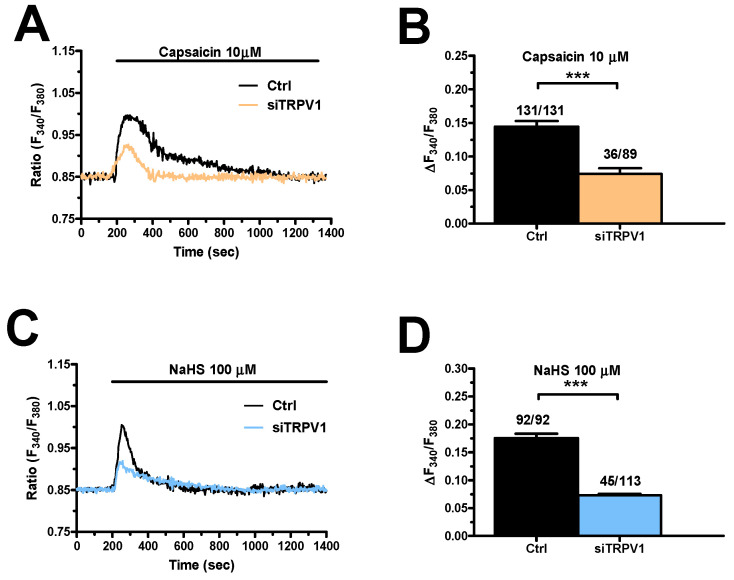
Genetic deletion of TRPV1 reduced NaHS-evoked extracellular Ca^2+^ influx in mCRC cells. (**A**), the increase in [Ca^2+^]_i_ evoked by capsaicin (10 µM) in mCRC cells transfected with a scrambled construct (Ctrl) was remarkably reduced in mCRC cells transfected with a selective siTRPV1. (**B**), mean ± SE of the percentage of the Ca^2+^ response to capsaicin under the designated treatments. (**C**), the increase in [Ca^2+^]_i_ evoked by NaHS (100 µM) in mCRC cells transfected with a scrambled construct (Ctrl) was remarkably reduced in mCRC cells transfected with a selective siTRPV1. (**D**), mean ± SE of the percentage of the Ca^2+^ response to NaHS under the designated treatments. Student’s *t*-test has been used for statistical comparison. In Panel B and D: *** *p* ≤ 0.001.

**Figure 11 cancers-12-03338-f011:**
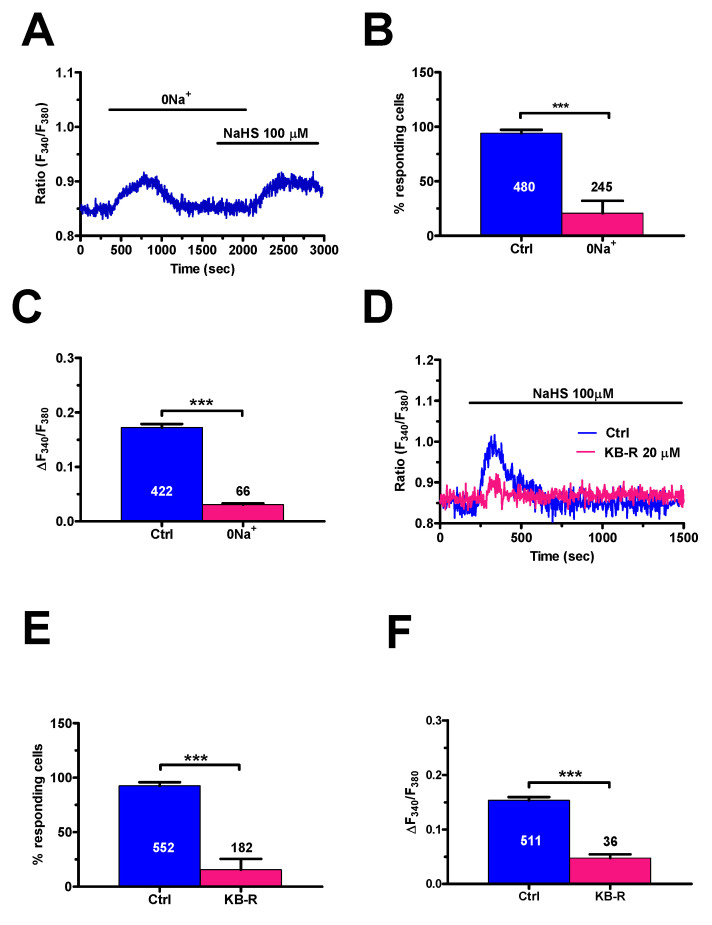
NaHS-evoked Ca^2+^ entry requires the reverse mode of NCX in mCRC cells. (**A**), the Ca^2+^ response to NaHS (100 µM) was prevented by removal of extracellular Na^+^ (0Na^+^) and resumed upon adding back Na^+^ to the perfusate. Please note that removal of extracellular Na^+^ resulted in a transient increase in [Ca^2+^]_i_ due to reversal of NCX activity. (**B**), mean ± SE of the percentage of cells presenting a discernible Ca^2+^ response to NaHS in the absence (Ctrl) and presence of extracellular Na^+^. (**C**), mean ± SE of the amplitude of NaHS-evoked Ca^2+^ influx in the presence (Ctrl) and absence of extracellular Na^+^. (**D**), pre-incubating mCRC cells with KB-R 7943 (KB-R; 20 µM, 20 min), a selective inhibitor of the forward-mode of NCX, abrogated the Ca^2+^ signal generated by NaHS (100 µM). (**E**), mean ± SE of the percentage of cells presenting a discernible Ca^2+^ response to NaHS in the absence (Ctrl) and presence of KB-R 7943 (KB-R). (**F**), mean ± SE the amplitude of NaHS-evoked Ca^2+^ influx in the absence (Ctrl) and presence of KB-R 7943 (KB-R). Student’s *t*-test has been used for statistical comparison. In Panels B, C, E and F, *** *p* ≤ 0.001.

**Figure 12 cancers-12-03338-f012:**
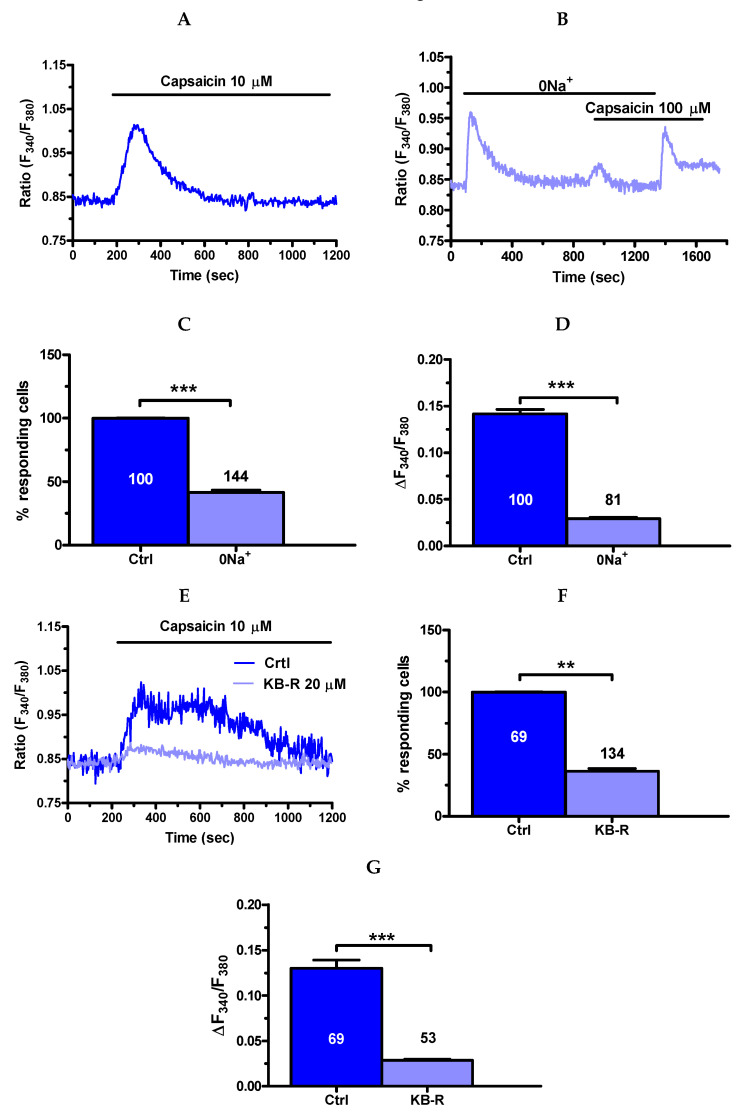
NCX contributes to capsaicin-induced extracellular Ca^2+^ entry in mCRC cells. (**A**), control Ca^2+^ response to capsaicin in the presence of extracellular Na^+^. (**B**), the Ca^2+^ response to capsaicin (10 µM) was dramatically reduced by removal of extracellular Na^+^ (0Na^+^) and resumed upon adding back Na^+^ to the perfusate. As described in Figure 11A, removal of extracellular Na^+^ resulted in a transient increase in [Ca^2+^]_i_ due to reversal of NCX activity. (**C**), mean ± SE of the percentage of cells presenting a discernible Ca^2+^ response to capsaicin in the presence (Ctrl) and absence of extracellular Na^+^. (**D**), mean ± SE of the amplitude of capsaicin-induced Ca^2+^ influx in the presence (Ctrl) and absence of extracellular Na^+^. (**E**), pre-incubating mCRC cells with KB-R 7943 (KB-R; 20 µM, 20 min) reduced the amplitude of the Ca^2+^ response capsaicin (10 µM). (**F**), mean ± SE of the percentage of cells presenting a discernible Ca^2+^ response to capsaicin in the absence (Ctrl) and presence of KB-R 7943 (KB-R). (**G**), mean ± SE of the amplitude of capsaicin-induced Ca^2+^ influx in the absence (Ctrl) and presence of KB-R 7943 (KB-R). Student’s *t*-test has been used for statistical comparison. In Panels C, D, F and G, *** *p* ≤ 0.001; ** *p* ≤ 0.01.

**Figure 13 cancers-12-03338-f013:**
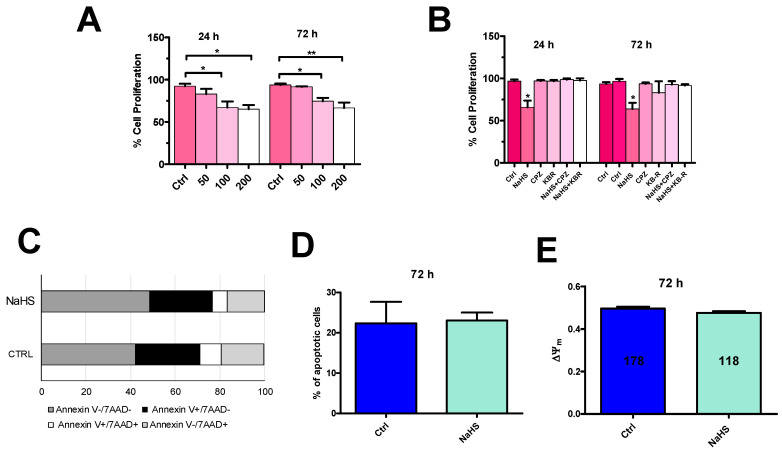
NaHS inhibits proliferation, but does not induce apoptosis, in mCRC cells. (**A**), incubation of mCRC cells for 24 h and for 72 h in presence of increasing concentrations of NaHS (50, 100 and 200 µM) reduced cell proliferation as determined by direct cell counting with the Trypan blue assay. (**B**), the anti-proliferative effect of NaHS (100 µM, 72 h) was prevented by pretreating mCRC cells with capsazepine (CPZ; 10 µM, 20 min) and KB-R 7943 (KB-R; 10 µM, 30 min). One-way ANOVA analysis followed by the post-hoc Dunnett’s test was used for Statistical comparison. In Panels A and B: ** *p* ≤ 0.001; * *p* ≤ 0.05. Each experiment was repeated three times. (**C**), Flow cytometry analysis of Annexin and 7-ADD staining. Representative cytogram of three independent experiments with the percentage of distribution of different population of mCRC cells in the absence (Ctrl) and in the presence of NaHS (100 µM, 72 h). Living cells (Annexin V−/7−ADD−) are represented in dark grey, early apoptotic cells (Annexin V+/7−ADD−) in black, late apoptotic/dead cells (Annexin V+/7−ADD+) in white, and dead cells (Annexin V+/7−ADD+) in light grey. (**D**), mean ± SE of the percentage of apoptotic mCRC cells in the absence (Ctrl) and presence of NaHS. (**E**), mean ± SE of ΔΨ_m_ in the absence (Ctrl) and presence of NaHS (100 µM, 72 h). ΔΨ_m_ was measured by evaluating tetramethylrhodamine, methyl ester (TMRM) fluorescence [67].

**Figure 14 cancers-12-03338-f014:**
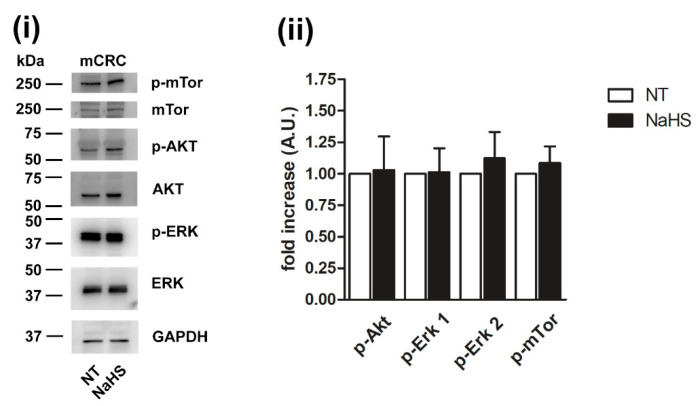
NaHS does not activate phosphorylation cascades in mCRC cells. Phosphorylation of the indicated selected signaling proteins in mCRC cells incubated with 100 µM NaHS for 24 h. Representative immunoblots with specific anti-phosphoprotein antibodies directed against the different substrates are reported (**i**). Akt, Erk and mTOR staining were used for equal loading control for corresponding specific phosphoprotein. GAPDH staining was used as equal loading control. Quantification of the results performed by densitometric scanning is reported in (**ii**), as fold increase (A.U.) of phosphorylation over basal (NT). Results are the mean ± SD of three different experiments.

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
