# Peer review of "Hydrogen Sulfide-Evoked Intracellular Ca2+ Signals in Primary Cultures of Metastatic Colorectal Cancer Cells"

_cancers, 2020, doi:10.3390/cancers12113338_

Round 1

Reviewer 1 Report

In this article entitled “Hydrogen sulphide-evoked intracellular Ca2+ signals in primary cultures of metastatic colorectal cancer cells”, Faris et al show that NaHS application induce calcium responses in human mCRC cells, and link them to TRPV1 and NCX activity. While the manuscript is generally well written and clearly presented, some extra information and experiments should still be added before it becomes suitable for publication.

Major comments:

While the authors present interesting data on the effect of NaHS in mCRC, the choice of this model is never justified. Is NaHS only eliciting responses in metastatic CRC, with no effect on normal or colorectal cancer cells originating from the primary tumor? In methods section, the authors specify that they had access to primary tumors, so why did they choose to focus on metastatic cells?

Are TRPV1 and NCX differentially expressed in normal and cancer cells? This information should be added to the manuscript, and discussed in regard to the therapeutic interest of NaHS application in CRC.

In Fig1A, please clarify if each trace is a single representative cell or the mean of different experiments.

The authors do not mention Store-Operated Channels (the main calcium entry pathway in non-excitable cells), or any TRP and VGCC channels as potential targets for NaHS. How can they completely rule out the involvement of any of these calcium channels in the potentially complex responses elicited by NaHS in CRC cells? Store depletion, even when barely detectable with Fura-2, can still lead to SOC activation. Moreover, while SB 366791 is currently considered as a specific inhibitor of TRPV1, confirmation of TRPV1 involvement in NaHS-induced calcium entry through silencing would further strengthen the authors conclusions.

Fig 7: the apparent molecular weight for the band presented by the authors as TRPV1 is around 60-65 kDa, a value rather different from its theoretical 90 kDa. Moreover, IF staining shows a TRPV1 signal co-localizing with tubulin, thus suggesting a mostly intracellular localization of the channel along cytoskeleton, with no clear signal detected at cell surface. Taken together, these results do not support the presence of functional TRPV1 channels in CRC cells membranes. As suggested before, confirmation of these results in cells devoid of TRPV1 (through silencing or KO) would go a long way in supporting the authors conclusions.

Fig 11: Confirmation of the effect of NaHs on cell proliferation by another method beyond simple cell counting (e.g. cell cycle analysis) could lead to the identification of subtle effects not visible here. Moreover, it seems difficult to evaluate proliferation in a population where 20% of the cells are undergoing apoptosis in control conditions. How can the authors explain such a high number of apoptotic cells? Were these values affected by passage number? How long were cells kept in culture? Can the authors please comment on how calcium levels and responses to NaHS would be affected by apoptosis in their model?

Reviewer 2 Report

Faris et al. showed that H2S promoted Ca2+ signaling pathway by stimulating TRPV1 Ca2+ channel, and TRPV1 activation also induced NCX (reverse mode)-mediated Ca2+ influx. The authors also showed that anti-proliferative effects of H2S in mCRC cells was not due to apoptosis. The experimental techniques and strategies are reliable, however, there are some critical concerns that need to be addressed.

Major concerns:

1. The authors showed that apoptosis and mitochondrial dysfunction were not involved in the promotion of H2S-induced cell proliferation. However, the underlying mechanisms remains unclear in this manuscript. Generally, elevation of intracellular Ca2+ induces cell proliferation or apoptosis, and TRPV1 can modulate both balance in cancerous cells. Therefore, in this study, it is essential to explain the underlying mechanism. The authors should investigate the TRPV1 (NCX)-induced modulation (activation / phosphorylation) of Ca2+-dependent signaling pathways like CaMK, MAPK-ERK, PI3K-AKT-mTOR, CN-NFAT, cyclin-dependent kinases, and so on.

Minor concerns:

1. In Fig. 1B, 3B. 5B, 6B, 8B, 9B, 9E, 10C, and 10F: Show the criteria for ‘responding cells’

2. Fig 1B, 1C, 6C, 8B, 8C: It is obscure which groups are statistically significant.

3. Fig. 11A, 11B, 11C: Show the examined experimental / cell numbers.

Reviewer 3 Report

This manuscript by Faras et al., investigates in vitro the effect of hydrogen sulphide as potential anti-cancer drug in primary cultures of metastatic colorectal cancer using a multidisciplinary approach. The results obtained suggest a mechanism in which the hydrogen sulphide mediated activation of TRPV1 channel induces extracellular Ca2+ entry and, in turn, suppresses cells proliferation by arresting the cell cycle in the S-phase.

Overall, this research is well written, and the content of this manuscript is of major interest. Nevertheless, the following issues (mainly minors) need to be addressed:

- Abstract: Line 35, please correct the typo “in in”. Lines 44-45: I would not finish the abstract with this sentence. Please rephrase

- Introduction: The authors performed all the experiments based on the concept that hydrogen sulphide is a specific and selective agonist (activator) of TRPV1. Is it true? I mean, it is possible the activation of other channels or receptors mediated by hydrogen sulphide? The authors should clarify this in the introduction.

- References in the text are wrong. References must be numbered in order of appearance in the text and they should be placed in square brackets.

- Line 77: ….promoting angiogenesis. A reference is needed

- Fig. 7A: The image of WB for TRPV1 you show in supplementary is better than the one you show in this figure. Please replace it

- Fig. 7B: Line 222-224: “The simultaneous staining of cell nuclei and microtubules reported in the images at higher magnification (Fig. 7B, panel ii) allowed to better appreciate the distribution of TRPV1 in the cell”. Are you sure? I can appreciate “red” in all the cell. For example, looking at this immunofluorescence image you can not know if TRPV1 is localized in the plasma membrane. I would not have used Tubulin.

- Discussion is too large and partially repeat results. As such, I suggest the author reduces this section to keep only the most important elements.

- Line 532: Please correct the typo “4.3[. Ca2+]i measurements”

- Methods, Chapter 4.4 and 4.5: The author must specify the catalog number of all the antibodies used: TRPV1, secondary anti-rabbit HRP, GAPDH, tubulin….

- Reference list is wrong: The reference list should include the full title, as recommended by the ACS style guide.

Author 1, A.B.; Author 2, C.D. Title of the article. Abbreviated Journal Name Year, Volume, page range

Round 2

Reviewer 1 Report

While I am fully satisfied with the extra-experiments performed by the authors, and the modifications they made to the manuscript, I still have two comments:

Figure S2: could the authors please specify what concentrations of siRNA they used? What was the transfection method? How long were the cells exposed to siRNA prior to the experiments? I did not find this information in the Methods section.

Figure S4 shows a clear effect of BTP2 (a non-specific SOC/TRPC inhibitor) on the NaSH-induced calcium entry. Indeed, calcium entry (that would be here better represented by the area under the curve rather than the peak amplitude) seems to be significantly reduced after BTP2 application. This result, if statistically significant, would therefore suggest the involvement of SOC and/or TRPC channels in the response to NaSH, even if TRPV1 remains a key actor of this response as shown by the authors. Can the authors rule out the possibility that at least part of the effects observed are due to a pool of TRPV1 channels localized in ER membrane that could induce SOCE in response to NaSH application?

Author Response

We are gratefully thankful for your comments on our manuscript entitled: “Hydrogen sulphide-evoked intracellular Ca2+signals in primary cultures of metastatic colorectal cancer cells” for publication as researchArticle inCancersWe truly believe that your comments significantly improved the qualityof our manuscript, which was amended by addressing the criticisms you raised.

Major comments:Figure S2: could the authors please specify what concentrations of siRNA they used? What was the transfection method? How long were the cells exposed to siRNA prior to the experiments? I did not find this information in the Methods section.

The Referee is fully right. Wedescribed the methods of gene silencing in Paragraph 4.6 of the Materials and methods.

Figure S4 shows a clear effect of BTP2 (a non-specific SOC/TRPC inhibitor) on the NaSH-inducedcalcium entry. Indeed, calcium entry (that would be here better represented by the area under the curve rather than the peak amplitude) seems to be significantly reduced after BTP2 application. This result, if statistically significant, would therefore suggest the involvement of SOC and/or TRPC channels in the response to NaSH, even if TRPV1 remains a key actor of this response as shown by the authors. Can the authors rule out the possibility that at least part of the effects observed are due to a pool of TRPV1 channels localized in ER membrane that could induce SOCE in response to NaSH application?

We thank the Referee for this valuable comment.We changed the trace in the graph. The point you raised is very important. However, we are sure that BTP-2 did not affect the Ca2+response to H2S. My postdoctoral fellow did not have the time to go through the complete analysisof the kinetics/area under the curveof -evoked Ca2+signals in the absence and in the presence of BTP-2, but (as in control cells)the Ca2+response was quite variable, ranging from transient to long-lasting Ca2+signals.So, we truly believe that, even though TRPV1 is inthe ER, it is unlikely to trigger SOCE under our conditions.

But the hypothesis you suggested couldbe true in another cellular model we are working on: endothelial colony forming cells. We have discovered that, in ECFCs, capsaicin evokes both intracellularCa2+releaseand extracellular Ca2+entry. Fura-2 imaging clearly shows ER-dependent endogenous Ca2+mobilization under 0Ca2+conditions (which we can never see in mCRC cells). In these days, as long as our departmentremains open(Covid-19 ishittinghard in Pavia),we are investigating the effect of BTP-2 on capsaicin-evoked Ca2+entry in ECFCs.

Once again,we truly thank you for the carefulevaluation of our manuscript and do hope that you will now regard our manuscript suitable for publication on this Special Issue of Cancers.

Reviewer 2 Report

The authors were adequately addressed in the revised manuscript. I have no more concerns.

Author Response

We thank the Referee for her/his nice comments and careful evalutation of the manuscript. We did appreciate this!

Reviewer 3 Report

The authors have addressed all my comments/suggestions. I found their responses quite satisfactory and the revised version has been much improved. I now recommend the paper for publication in Cancers

Author Response

(The authors gave the same response as above.)
